# The Role of Silica Nanoparticles in Promoting the Germination of Tomato (*Solanum lycopersicum*) Seeds

**DOI:** 10.3390/nano13142110

**Published:** 2023-07-19

**Authors:** Anca Awal Sembada, Shinya Maki, Ahmad Faizal, Toshiyuki Fukuhara, Takeshi Suzuki, I. Wuled Lenggoro

**Affiliations:** 1Chemical Engineering Program, Graduate School of Engineering, Tokyo University of Agriculture and Technology, Koganei 184-8588, Tokyo, Japan; anca@st.go.tuat.ac.jp; 2Department of Science of Technology Innovation, Nagaoka University of Technology, Nagaoka 940-2188, Niigata, Japan; 3Plant Science and Biotechnology Research Group, School of Life Sciences and Technology, Institut Teknologi Bandung, Bandung 40132, Indonesia; afaizal@itb.ac.id; 4Department of Applied Biological Science, Graduate School of Agriculture, Tokyo University of Agriculture and Technology, Fuchu 183-8509, Tokyo, Japan; 5Graduate School of Bio-Applications & Systems Engineering, Tokyo University of Agriculture and Technology, Koganei 184-8588, Tokyo, Japan; 6Department of Applied Physics and Chemical Engineering, Graduate School of Engineering, Tokyo University of Agriculture and Technology (TUAT), Koganei 184-8588, Tokyo, Japan

**Keywords:** silicon dioxide, colloid, plants, growth, *Bacillus* sp.

## Abstract

The addition of nanoparticles has been reported to be an effective strategy for enhancing seed germination, but the underlying mechanisms whereby this occurs are unclear. In the present study, we added silica nanoparticles (SiNPs) to an aqueous growth medium in which tomato seeds were germinated. We examined the effects of SiNPs on growth and possible mechanisms of action. SiNPs had a diameter of 10–17 nm and 110–120 nm. SiNPs shortened the mean germination time from 5.24 ± 0.29 days to 4.64 ± 0.29 days. Seedling vigor, measured by criteria including length and weight, was also improved compared to the control condition. The presence of SiNPs in the seedlings was assessed using an X-ray fluorescence spectrometer. The nanoparticles may have promoted germination by enhancing water imbibition by the seeds or altering the external microenvironment. Scanning electron microscopy revealed changes in the seed coat during germination, many of which were only observed in the presence of nanoparticles. Soil bacteria affect germination; specifically, *Bacillus* sp. may promote germination. The number of *Bacillus* sp. changed in the germination medium with SiNPs compared to the control. This suggested that these bacteria could interact with SiNPs to promote germination.

## 1. Introduction

Treatment of seeds to enhance germination is an important component of agricultural intensification. Vigorous seed germination and robust seedling growth provide the foundation for a healthy plant [1] and impact yield and productivity. Cultivation methods that promote seed germination enable the radicle to access nutrients and to become situated in the soil environment early in germination [2].

Nanoparticles (less than 100 nm in diameter) have been reported to hasten seed germination [3]. These particles may be soluble or insoluble in suspension. Both have been used in agriculture at many stages of plant development, including in seed germination [4,5] and mature plants [6]. Insoluble nanoparticles can be dropped in a suspension on the media (e.g., cotton puffs or filter paper) that contain the seeds [3]. Nanoparticles have also been used to prime seeds by coating the seed surface [7]. When tomato (*Lycopersicum esculentum*) seeds are placed on filter paper with colloidal SiNPs (12 nm), the germination percentage increases, time to germination decreases, and vigor is enhanced (as measured by dry and fresh weight) [3]. The germination of common bean (*Phaseoulus vulgaris*) seeds on filter paper has also been shown to be enhanced when SiNPs (10 nm) were applied [8]. Feizi et al. applied suspensions of TiO_2_ particles (21 nm) to wheat seeds (*Triticum aestivum* L. var. Pishtaz) on moistened paper and noted that germination was promoted [9]. Camphor tree seed germination was also promoted by the application of TiO_2_-NPs [10]. CeO_2_-NPs (25 and 50 nm) could promote both monocot (*Holcus lanatus*) and dicot (*Diplotaxis tenuifolia*) seed germination [11]. Among different types of nanoparticles, silica has attracted attention due to its inertness and stability. While the studies mentioned above provide evidence that nanoparticles can enhance seed germination, the mechanism of these effects remains largely undiscussed. One study proposed that nanoparticles increase the α-amylase and soluble sugar content of the seeds [12]. Nanoparticles were also found to maintain the production of reactive oxygen species (ROS) in treated germinating seeds compared to the untreated seeds that showed increased [7] or decreased ROS levels [12].

Little information and discussion are available on the transport of nanoparticles during the seed germination process. On the other hand, nanoparticles of various sizes and wettability exist, and are often applied to seeds or growth media at different concentrations. Therefore, their role in seed germination is expected to be variable. Another important consideration is the interaction between microorganisms in the microenvironment surrounding the seeds and the SiNPs. *Bacillus* sp. is a model bacterium for studying the interactions of microorganisms and minerals [13], and its interaction with silica in the soil has been considered in the study of seed germination and development [14]. It is known, for example, that *B. subtilis* promotes the germination of wheat seeds [15]. However, the nature of the interaction between SiNPs and *Bacillus* sp. in the process of seed germination has not been studied. Different NPs sizes also play an important role in influencing germination. A study conducted by Alkhatib et al. reported that the effect of NPs was highly dependent on size and concentration [16]. In the present study, water insoluble SiNPs (with two sizes) were used to further investigate the mechanism whereby nanoparticles promote tomato seed germination. The transport of nanoparticles and their interactions with microorganisms during seed germination were also investigated.

## 2. Materials and Methods

### 2.1. Seeds Viability Test

Tomato seeds (*Solanum lycopersium* var. Momotaro) were obtained from Takii & Co., Ltd., Kyoto, Japan. Seeds were washed with distilled water to clean the surfaces [17] and soaked in 50 mL of distilled water in a 100 mL glass for 10 min. Seeds that sank were deemed viable [18]. We selected 225 viable seeds for the germination test, which consisted of nine treatments, each with 25 seeds.

### 2.2. Preparation of Nanoparticle Suspensions

Colloidal SiNPs, (Nissan Chemical Corporation, Tokyo, Japan) either Snowtex-S with diameters of 8–11 nm or Snowtex-ZL with diameters of 70–100 nm, were used. These aqueous nanoparticles are amorphous, spherical [19], negatively charged [20], and stable at an alkaline pH of 9.5–10.5 without added organic surfactants [21]. Nanoparticles were used as received and diluted to the specified concentration for further investigation. Five different concentrations of nanoparticles were used: 0 (control), 10, 100, 1000, and 2000 mg/L [3]. The final pH of the suspensions was adjusted to pH 9.5–10.5 with NaOH (No. 198-13765, purity 97%, Fujifilm Wako Pure Chemical Co., Ltd., Osaka, Japan).

### 2.3. Dispersity and Size Distribution of Nanoparticle Suspensions

To determine the zeta potential and particle size distribution of the suspension, we used a dynamic light-scattering zeta sizer (DLS Zetasizer Nano-ZS, Malvern Panalytical Ltd., London, UK). To measure the distribution of the particle size, 4 mL of suspension was placed in a 5 mL plastic cuvette, and the cuvette was placed in the cell area of the zeta-sizer device. We used a folded capillary cell for the zeta potential measurements, as follows: 1 mL of the suspension was put into the capillary cell and the cell was placed in the cell area of the zeta sizer. Each treatment was measured three times.

### 2.4. Seed Germination

To assess germination, five tomato seeds were placed in a Petri dish (9 cm diameter) that had been layered with a 0.8 g cotton puff. Twenty mL of nanoparticle suspensions was added [22]. We tested all five concentrations and both sizes of particles, performing five replicates for each treatment (45 total). Dishes were sealed with paraffin tape and placed in a closed cupboard box (40.5 cm × 34.5 cm × 20 cm) for 8 days in the dark [3]. On the 2nd, 4th, and 6th days of germination, 5 mL of water was added to each Petri dish.

### 2.5. Germination Parameters

Seeds were considered germinated when the radicle was at least 2 mm long [3]. We used several parameters to evaluate germination. The first parameter was the seed germination percentage (SGP), or the percentage of seeds that had germinated in each treatment on a particular day. SGP was calculated with the following formula [23]:(1)$$\mathrm{SGP}=\frac{\mathrm{number}\;\mathrm{of}\;\mathrm{seeds}\;\mathrm{germinated}}{\mathrm{total}\;\mathrm{number}\;\mathrm{of}\;\mathrm{seeds}}\;\times \;100\%$$

The second parameter was mean germination time (MGT), a measure of how quickly seeds germinated. Lower values indicated rapid germination. Mean germination time was calculated with the following formula [3,24]:(2)$$\mathrm{MGT}=\frac{\Sigma \;n.d}{\Sigma \;n}$$
where d is the number of days needed for seeds to germinate and n is the number of seeds that germinated on day d. The third parameter was the germination index (GI), which couples the percentage of germination and speed of germination. Higher GI values indicate both increased percentages of germination and a shorter germination period. GI was calculated using the following formula [24]:(3)$$\mathrm{GI}=\left(8\;\times \;n_{1}\right)+\left(7\;\times \;n_{2}\right)+\left(6\;\times \;n_{3}\right)+\ldots +\left(1\;\times \;n_{8}\right)$$
where n_1_, n_2_, …, n_8_ are the number of seeds that germinated on the first, second, and each subsequent day until the 8th day, and the first number in each term (8, 7, …, and 1) is the weight given to the number of seeds that germinated on the first, second, and each subsequent day [24]. The fourth parameter was the coefficient of velocity of germination (CVG), which represents the rapidity of germination. It was calculated using the following formula [24,25]:(4)$$\mathrm{CVG}=\frac{\left(n_{1}+n_{2}+\ldots +n_{i}\right)}{\left(n_{1}T_{1}+n_{2}T_{2}+\ldots +n_{i}T_{i}\right)}\times \;100$$
where n is the number of seeds that germinated each day, T is the number of days since germination was initiated, and i is the number of days needed for germination until harvesting. The fifth parameter was a set of vigor indices (VI). These are indicators of seedling traits, such as root and shoot length (LENGTH), fresh weight (FW), and dry weight (DW). The VI values were calculated according to the following formulas [3,25]:(5)$${\mathrm{VI}}_{\mathrm{LENGTH}}=\mathrm{SGP}\;\times \;\mathrm{mean}\;\mathrm{of}\;\mathrm{seedling}\;\mathrm{length}\;(\mathrm{root}+\mathrm{shoot})$$
(6)$${\mathrm{VI}}_{\mathrm{FW}}=\mathrm{SGP}\;\times \;\mathrm{mean}\;\mathrm{of}\;\mathrm{seedling}\;\mathrm{fresh}\;\mathrm{weight}$$
(7)$${\mathrm{VI}}_{\mathrm{DW}}=\mathrm{SGP}\;\times \;\mathrm{mean}\;\mathrm{of}\;\mathrm{seedling}\;\mathrm{dry}\;\mathrm{weight}$$

### 2.6. Harvesting and Measurement of Seedling Weight

Seedlings were harvested on the last day of germination (day 8). Roots and stems from seedlings were carefully separated from the cotton puff to keep them intact. The length and fresh weight of the seedlings were measured. Seedlings were rinsed with water and dried in an oven for 48 h at 50 °C to obtain the dry weight of the biomass [26].

### 2.7. Elemental Analysis of Seedlings

Eight-day-old dried seedling samples were weighed to 0.001 g and put into a mechanical press machine under 100 kgf/cm^2^ pressure to form pellets (1 cm diameter). Pellets were placed in the sample holder for an X-ray fluorescence (XRF) spectrometer (JSX-3100RII, JEOL Ltd., Tokyo, Japan). The XRF conditions were tube voltage = 30 kV, collimator = 1, and live-time value = 100 s.

### 2.8. Imaging of Seed Surface

The morphology of untreated seeds and seeds that had been treated with SiNPs was examined using a scanning electron microscope (SEM; JSM-6510, JEOL Ltd., Tokyo, Japan) on days 0, 1, 2, and 3 after the start of germination [27]. Before SEM examination, seeds were dried in an oven at 30 °C for 24 h and coated with platinum (Pt) using an ion sputtering device (Fine Coat Ion Sputter JFC-1100, JEOL Ltd., Tokyo, Japan). The coating treatment caused the seeds to become conductive, which made them easier to visualize later. Coated seeds were then set in the stage holder of the SEM device under vacuum. The SEM conditions were as follows: voltage = 2 kV, working distance (WD) = 11 mm, and strain sensors (SS) = 60.

### 2.9. Isolation and Quantification of Bacillus sp. from Germination Media

*Bacillus* sp. form rod-shaped spores. This distinctive spore morphology facilitated their isolation and quantification. Bacteria was isolated from germination media (water and SiNPs) on the first and last days of germination. To assess bacterial counts, the collected germination media were diluted with water (1:100). The diluted media were incubated at 4 °C for 24 h and then incubated at 80 °C for 30 min to eliminate the non-sporulating microorganisms [28]. A total of 100 µL of the heat-treated germination media was distributed over nutrient broth agar by spreading. Plates were incubated at 37 °C for 48 h. We reasoned that the spores that survived would germinate and form colonies, which could be quantified as colony forming units (CFU). The following equation was used to calculate CFU/mL:(8)$$\mathrm{CFU}/\mathrm{mL}=\frac{\mathrm{number}\;\mathrm{of}\;\mathrm{colonies}\;\times \;\mathrm{total}\;\mathrm{dilution}\;\mathrm{factor}\;}{\mathrm{volume}\;\mathrm{of}\;\mathrm{cultured}\;\mathrm{plate}\;\mathrm{in}\;\mathrm{mL}}$$

### 2.10. Statistical Analysis

Data were evaluated with descriptive statistics (mean and population standard deviation) in Microsoft Excel. IBM SPSS Statistics was used to perform the two-way analysis of variance (ANOVA), Duncan’s multiple range test (DMRT), and Tukey’s range test. Differences at *p* ≤ 0.05 were considered statistically significant.

## 3. Results

### 3.1. Characteristics of Nanoparticles

We measured the zeta potential and the size distribution of the suspended nanoparticles used in this study, as shown in Table 1. All measurements of zeta potentials were greater than or approximately equal to −30 mV at pH 9.5–10.5, indicating that the suspensions were electrostatically stable [29] and could handle strong electrostatic repulsion [30]. The particle size distributions, measured on a DLS zeta sizer, were approximately 10–17 or 110–120 nm (Table 1). These will be referred to as small and large SiNPs hereafter. It should be noted that the manufacturer reported that the distributions were 8–10 nm or 70–100 nm when observed under SEM. The particle size determined by SEM is often smaller than that measured by DLS due to the difference in measurement principles of the two methods; particles analyzed by SEM are in the dry state, while those assessed with DLS are in a hydrodynamic state [4].

### 3.2. Seed Germination and Seedling Growth in the Presence of Nanoparticles

SGPs are shown in Figure 1. All seeds in all treatments germinated by the last day of the eight-day experiment, but the time required for germination varied between treatment groups. Other parameters of germination, specifically MGT, GI, and CVG, are shown in Table 2. Mean germination time was significantly different (*p* ≤ 0.05) in all treatment groups compared to the untreated control group. The control seeds germinated in 5.24 ± 0.29 days, and the shortest germination time, 4.64 ± 0.29 days, was observed when seeds were germinated in a suspension of small SiNPs at a concentration of 1000 mg/L. The germination time of seedlings in a suspension of large SiNPs at a concentration of 100 mg/L was also less than the control germination time (4.68 ± 0.27 days). Thus, SiNPs hasten the germination of tomatoes and perhaps other species.

Values for GI and CVG for all treatment groups were significantly different compared to the untreated group (*p* ≤ 0.05). The greatest GI, a parameter that includes both the percentage of germinated seeds and the speed of germination, was observed in seeds germinated in suspensions of small SiNPs at a concentration of 1000 mg/L. The GI for these seeds was 21.8 ± 1.47 and the CVG was 21.64 ± 1.34 compared to control values of 18.8 ± 1.47 and 19.14 ± 1.01, respectively. Seeds germinated in 100 mg/L suspensions of large SiNPs also had a greater GI (21.6 ± 1.36) and CVG (21.44 ± 1.23) than the control seeds. Concentrations of 100 to 1000 mg/L for both sizes of particles were the best concentrations to use among others. A study conducted by Lin and Xing using zinc nanoparticles demonstrated that seed germination was inhibited at a nanoparticle concentration of 2000 mg/L [31]. A decrease in germination parameters was also observed when concentrations of 10–14 g/L SiNPs were used in tomato seed germination [3]. Therefore, the concentration of nanoparticle suspension used can affect the resulting germination performance.

The smaller nanoparticles, in general, were more effective at promoting germination than the larger particles. This may be due to the greater contact area between the small particles and the seed (surface) than between the larger particles and the seed. This finding may also be related to the hydrophilic nature of silica [32,33]. Increasing the contact surface area between SiNPs and seeds could enlarge the hydrophilic space around the seeds and enhance imbibition, thus promoting earlier germination. This hypothesis is compatible with the visualization of seed germination using SEM, which showed the surface of the seed in contact with the cotton puff and SiNP suspension (Figure 2). 

The seeds in the germination assay had reached the young seedling stage by the end of the assay. We investigated the role of SiNPs in the growth of these seedlings as well as in germination. The VIs we used were based on length, seedling FW, and seedling DW (Table 3). There were significant differences in the VIs in all treatment groups compared to the control group (*p* ≤ 0.05). The greatest VIs were detected when seeds and seedlings were germinated and grown in suspensions of small SiNPs at 1000 mg/L or large SiNPs at 100 mg/L. The VI_LENGTH_ values for these treatments were 1117.2 ± 81.54 and 1102 ± 121.36, respectively. The VI_FW_ and VI_DW_ for the seedlings that had been germinated in small SiNPs at 1000 mg/L were 2.52 ± 0.11 and 0.22 ± 0.01, respectively. For the large SiNPs at 100 mg/L concentration, VI_FW_ and VI_DW_ were 2.52 ± 0.11 and 0.23 ± 0.01, respectively. Our finding that germinating and growing seedlings in SiNPs increased vigor is consistent with the results of Siddiqui and Al-Whaibi [3], Alsaeedi et al. [8], and Ivani et al. [34], who also used cotton or cellulose filter paper for germination. They also concluded that nanoparticles not only accelerated germination but also improved seedling vigor after germination.

### 3.3. Elemental Analysis of Seedlings during Germination

Elemental analysis was performed on the seedlings on the last day of germination to track the presence of Si (of SiNPs) in the tissue. Eight elements, including magnesium (Mg), phosphorus (P), sulfur (S), potassium (K), calcium (Ca), manganese (Mn), iron (Fe), and zinc (Zn), were detected in all seedlings from all treatment groups. Si was not detected in either the control seeds or all treated seeds. We prepared a heat map to visualize the element levels in each treatment group (Figure 3). Thus, it can be concluded that the concentration of silica used in the present study is neither toxic nor at a level that affects the internal elements or physiological functions of the seedlings. 

### 3.4. Bacillus sp. Levels in Germination Media before and after Germination

Another possible mechanism of SiNP-induced germination enhancement could be the presence of bacteria that could help to hasten the germination by altering the seed morphology. Several studies have demonstrated that bacteria aid seed germination. Widnyana and Javandira showed that *Bacillus* sp. were more effective in stimulating tomato seed germination than *Pseudomonas* sp. [35]. It is perhaps not surprising that *Bacillus* sp. is a soil bacterium that is antagonistic to pathogenic fungi, and also enhances seed germination [36]. Furthermore, *Bacillus* sp. interacts with silicate minerals and can solubilize silicate in the environment (soil/water systems) [37,38]. Taken together, these observations support the notion that *Bacillus* sp. in combination with SiNP treatment can accelerate seed germination. There are currently no studies that discuss the interactions or combined beneficial effects of *Bacillus* sp. and nanoparticles on seed germination.

In the present study, *Bacillus* sp. were isolated from the germination media (water and SiNPs). The number of *Bacillus* sp., expressed as colony-forming-unit (CFU)/mL, was determined on day 1 and day 8 of germination (Figure 4). There were significantly more *Bacillus* sp. (*p* ≤ 0.05) in the germination medium on the first day of germination compared to media from the control group. The number of *Bacillus* sp. in the control was relatively very low. The presence of *Bacillus* sp. on the first day may have helped the seeds to germinate more rapidly. There was no significant difference in the number of *Bacillus* sp. on the last day of germination between the control and SiNP-treated groups. This was likely because the growth of *Bacillus* sp. had reached the stationary phase during which there are equal numbers of regenerating and dying bacteria.

## 4. Discussion

Increasing the concentration of SiNPs caused an increase in the value of the zeta potential and particle size distributions measured (Table 1). When nanoparticles were dispersed in a liquid medium, they acquired a surface charge due to the adsorption of ions or functional groups [39]. This surface charge determined the zeta potential. If the surface charge was negative, the zeta potential would be negative, and the nanoparticles would repel each other. The concentration of nanoparticles could affect the zeta potential through factors such as particle–particle interactions or the availability of ions in the suspension [40]. At low concentrations, the electrostatic interactions between nanoparticles might be weaker, leading to less negatively charged (zeta potential) values. Dilution of the nanoparticle suspension could potentially result in changes in the particle size distribution [41]. Dilution could disrupt attractive interactions between nanoparticles.

The measured germination parameters were improved when treated with SiNPs. This finding is supported by the observation of Siddiqui and Al-Whaibi, who reported shorter germination times when tomato seeds were placed on filter paper pre-treated with 12 nm particles at concentrations of 2–14 g/L [3]. Another study, conducted by Alsaeedi et al., showed that the germination time of common bean seeds on moistened paper with 10 nm particles at a concentration of 100–400 mg/L was also less than that of control seeds [8].

The concentration of nanoparticles has been found to have a more pronounced effect on germination compared to the size of nanoparticles as shown by some of the germination parameters in Table 2 and Table 3. For example, a study reported that the particle size of silver nanoparticles had no significant effect on lettuce germination, but the concentration did [42]. Higher nanoparticle concentrations corresponded to an increased total surface area of nanoparticles available for interactions with seeds. This larger surface area provided more opportunities for chemical reactions, adsorption, and physical interactions with seed surfaces, potentially affecting germination-related processes [43]. While nanoparticle size could influence their physicochemical properties and interactions with seeds, the concentration often superseded size effects in terms of their impact on germination. However, it is worth noting that specific responses could still vary depending on the nanoparticle type, seed species, and experimental conditions.

Neither of the studies above evaluated the effect of pH on germination. In this study, alkaline pH was used at the time of germination. Laghmouchi et al. found that *Origanum compactum* seeds germinated as well at alkaline pH (pH 9) as they did at pH 7, while in acidic conditions (pH 2 and pH 3.5), germination was severely reduced [44]. Li et al. indicated that the germination of smooth cordgrass (*Spartina alterniflora*) was affected only slightly by pH that varied from pH 6.7 to 10.7 [45]. There were no notable adverse effects of using alkaline pH (around pH 9–10) during the germination process.

The smaller nanoparticles, in general, were more effective at promoting germination than the larger particles. This may be due to the greater contact area between the small particles and the seed (surface) than between the larger particles and the seed. This finding may also be related to the hydrophilic nature of silica. Increasing the contact surface area between SiNPs and seeds could enlarge the hydrophilic space around the seeds and enhance imbibition, thus promoting earlier germination. This hypothesis is compatible with the visualization of seed germination using SEM, which showed the surface of the seed in contact with the cotton puff and SiNP suspension (Figure 2). Seeds surrounded by nanoparticles tended to have “fewer” hairs on the testa than the control seeds after 24 h of germination. The process of testa rupture on the surface of the seed is the first stage of germination, which is characterized by a protrusion of the radicle [46]. The testa rupture causes the surface of the testa to become smoother due to the degradation of pectin, which is regulated by abscisic acid (ABA) and 12-oxo-phytodienoic acid (OPDA) [47]. Nanoparticle-coated seeds had germinated within 48 h, which was earlier than the control seeds, as indicated by the presence of radicles. In both treated and untreated seeds, radicles appeared after 72 h of germination. Thus, SiNPs result in smoother testa surfaces, which in turn resulted in earlier germination. Testa structure is an important characteristic for seed survival and germination [48]. Important characteristics of the testa include the height of the epidermis, the thickness of the scleral parenchyma [48], and the structure of the epidermal hairs [49]. The presence of a hairy structure on tomato seeds could become a site for infestation of some particles such as oospores (21.4–45.3 µm in size) from *Phytophthora* sp. and this condition could affect the physiology of the seed [50,51]. Something similar could also happen with the presence of SiNPs enclosed between the hairy parts of seed coat, which could later affect germination.

Internal elements in plants (Figure 3) can be indicators of the influence of the surrounding environment on plant growth. Every environmental change that occurs can change the composition and abundance of these elements [52]. Lin and Xing concluded that ZnO nanoparticles at an initial concentration of 2000 mg/L were toxic [31]. Nanoparticles might be toxic for two reasons. They might release toxic ions (if they are water-soluble), or there might be detrimental physical effects associated with the particles themselves. Physical effects may be associated with stress or harmful stimuli due to the nature of the particle surface, or the size and shape of the particles [31].

The absence of silica in the seedlings in all treatment groups (Figure 3) indicated that the silica concentration in the seedlings was probably below the resolution capacity of the XRF spectrometer (<10 ppm). Silica may only be present on the outside of the seed or in the area around the seed coat (testa). The presence of silica on the outside of the seed coat may have made the conditions around the seeds more hydrophilic and possibly attracted more water to the surface of the seed. This would accelerate the germination process and increase vigor. This is supported by observations from the SEM (Figure 2), which showed that the surface of the seeds treated with SiNPs became smoother and initiated earlier germination.

*Bacillus* sp. may improve germination in the presence of SiNPs. *Bacillus* sp. are plant growth-promoting rhizobacteria (PGPR), as their presence in the soil improves plant growth and helps plants resist environmental stress [53]. Thus, the changes in seed morphology in response to SiNPs could be partially due to increased levels of *Bacillus* sp. (Figure 4). A study conducted by Santos et al. showed that the presence of bacterial communities from the time of germination influences the morphology of maize seedlings [54].

We also conducted a follow-up study on the effect of the presence of *Bacillus* sp. and SiNPs during seed germination (data not shown). The study was carried out by sterilizing the seed surface with 70% ethyl alcohol [55] and sterilizing the cotton puff and SiNPs suspensions by autoclaving them at 121 °C and 15 psi for 30 min [26]. This step was carried out to eliminate the possibility of contamination by *Bacillus* sp. or other bacteria that may be present during germination. The results obtained showed that the germination profile between SiNP treatments was not significantly different. This supported the hypothesis that *Bacillus* sp. and SiNPs may have a synergistic effect on increasing germination in tomatoes. A study conducted by Haider et al. also demonstrated the synergistic application of *B. subtilis* and copper oxide nanoparticles (CuONPs, size < 50 nm) to improve the wheat (*T. aestivum*) and corn (*Zea mays*) seed germination, which was indicated by increased root length, shoot length, and plant biomass weight [56].

*Bacillus* sp., which helps seed germination, is possibly derived from the air surrounding the microenvironment of the seeds. In a study by Watanabe et al., *Bacillus* sp. was one genus of bacteria shown to frequently exist in the air in Japanese residential areas [57]. This allows *Bacillus* sp. to be acquired via the air even to the periphery of the seeds. As noted above, *Bacillus* sp. has been reported as silicate solubilizing bacteria; therefore, the presence of silica materials around seeds may attract more *Bacillus* sp. compared to the seed microenvironments that are not enriched in silica particles (Figure 4). *Bacillus* sp. is also capable of producing phytohormones. The study conducted by Khedher et al. showed that B. *subtilis* V26 could produce indole acetic acid (IAA), which plays a role in root and shoot cell division and elongation [58]. B. *subtilis* was also proven to have a positive effect on germination with seed priming of two different wheat (*T. aestivum* L.) cultivars, Ekada70 (E70) and Salavat Yulaev (SY) [59].

The observed germination times for a particular species of seed exhibited variability within a certain range. Even within a single species, there could be genetic variability among seeds. Different individuals within the same species might have slight genetic variations, leading to variations in germination characteristics, including the time it took for seeds to germinate [60]. This genetic diversity could contribute to the inconsistent germination times observed. To address this issue, we can take measures to minimize variability by using standardized protocols, maintaining consistent environmental conditions, and carefully selecting and handling seeds [61]. Conducting multiple replicates and statistical analyses could help identify trends and patterns within the variability and provide a clearer understanding of the average germination time for a particular species [62]. 

Future research on seed germination using nanoparticles should focus on various aspects to further our understanding of their effects. Investigating the dose–response relationships of nanoparticles in seed germination can provide insights into their toxicity thresholds or optimal concentrations. This study area can help establish guidelines for safe nanoparticle application in agriculture and highlight potential risks associated with high concentrations. Understanding the underlying mechanisms through which nanoparticles influence seed germination is crucial. Future research can focus on elucidating the cellular and molecular processes affected by nanoparticles, such as hormone regulation, gene expression, oxidative stress, and nutrient uptake. Advanced techniques like transcriptomics, proteomics, and metabolomics can provide valuable data in this regard. Nanoparticles can interact with the microbial communities associated with seeds, influencing seed germination outcomes. Future research can explore the effects of nanoparticles on the composition and activity of seed-associated microbiota and how these interactions impact germination. This area of study can shed light on the potential indirect effects of nanoparticles on seed germination through their influence on the microbiome. 

## 5. Conclusions

In this study, we investigated the effects of water insoluble silica nanoparticles (SiNPs) on the germination of tomato seeds in an aquatic environment. We found that SiNPs enhanced seed germination by reducing the time and increasing the rate of germination. SiNPs also improved the vigor of young seedlings, as indicated by their length and weight. The smaller size of SiNPs (10–17 nm) was more effective than the larger size (110–120 nm) in promoting seed germination and growth. We observed that the samples grown with SiNPs displayed a reduced number of hairs on the seed surface. We also examined the interactions of SiNPs with *Bacillus* sp. during seed germination. *Bacillus* sp. may have a positive effect on seed germination by producing plant growth-promoting substances. These results suggest that SiNPs can be used as an eco-friendly strategy to improve seed germination in aquatic environments.

## Figures and Tables

**Figure 1 nanomaterials-13-02110-f001:**
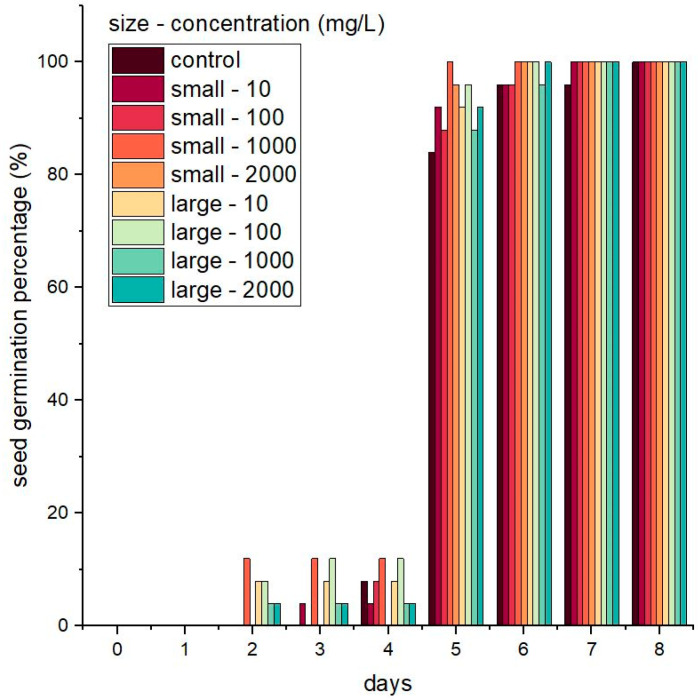
Germination of tomato seeds in the presence of different sizes of nanoparticles (small and large SiNPs) at different concentrations (0–2000 mg/L).

**Figure 2 nanomaterials-13-02110-f002:**
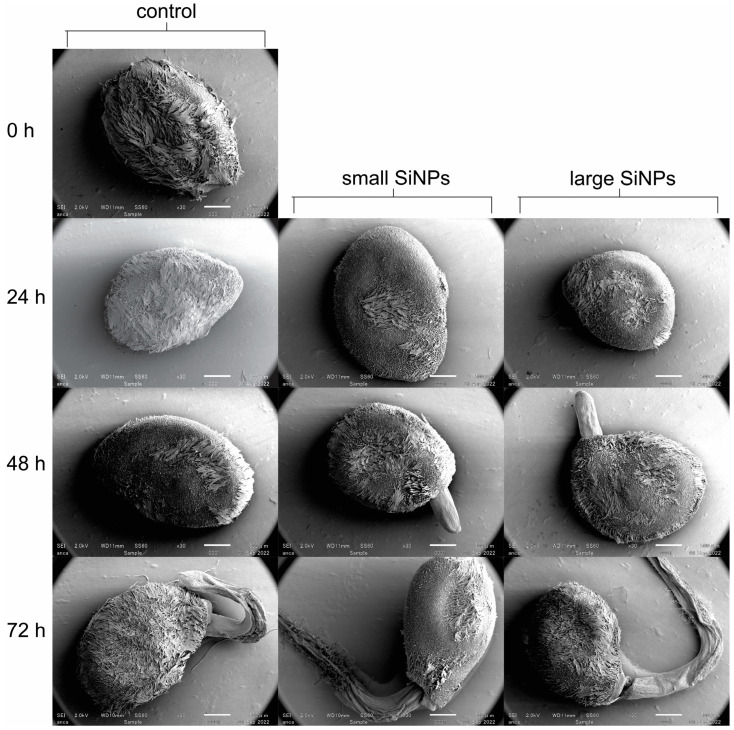
Scanning electron microscopic (SEM) images of the surface of tomato seeds after treatment with suspensions of small or large SiNPs for the indicated times. Images are of representative seeds from each treatment group.

**Figure 3 nanomaterials-13-02110-f003:**
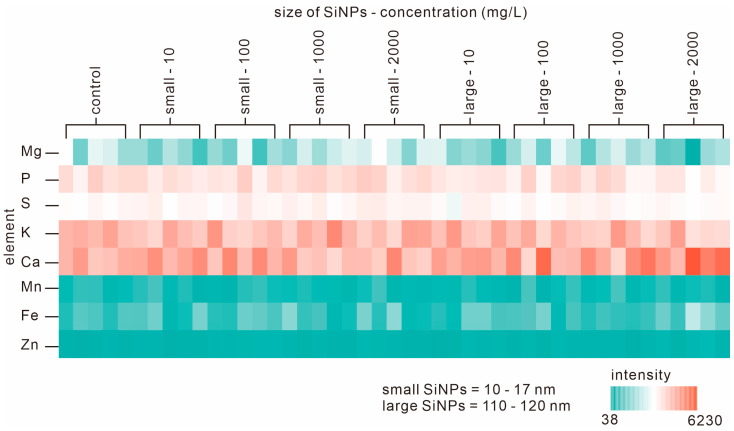
Heat map of elements detected in tomato seedlings that had been germinated and grown in suspensions of SiNPs of different sizes and at different concentrations.

**Figure 4 nanomaterials-13-02110-f004:**
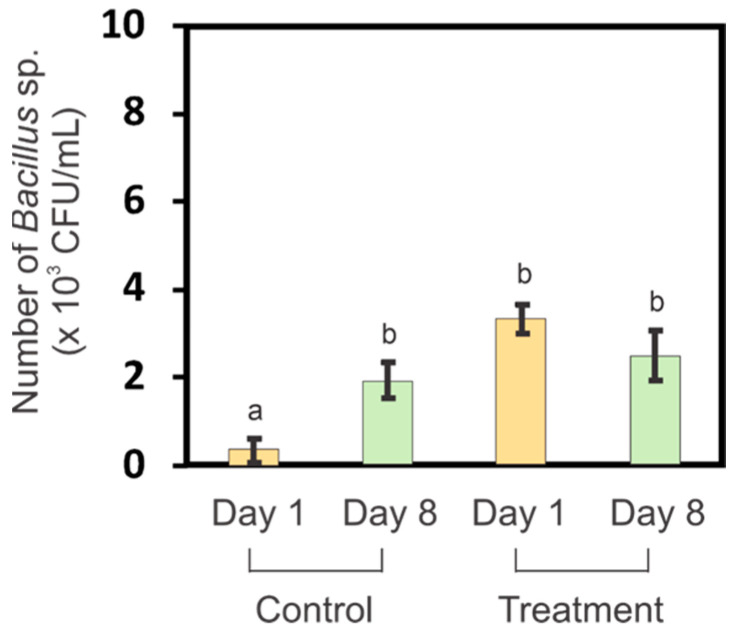
Number of *Bacillus* sp. colonies in germination media in which tomato seeds were germinated and grown for eight days. Germination media had either no SiNPs (control) or contained small SiNPs at 1000 mg/mL (treatment). Different letters indicate significant differences (*p* ≤ 0.05) according to Tukey tests.

**Table 1 nanomaterials-13-02110-t001:** Zeta potential and particle size distribution of silica nanoparticles. Different letters indicate significant differences (*p* ≤ 0.05) according to the Duncan test.

Silica Type	Concentration (mg/L)	Zeta Potential (mV)	pH	Particle Size Distribution (nm)
Snowtex-S (small SiNPs)	10	−29.3 ± 0.65 ^a^	9.94 ± 0.04	10.23 ± 0.21 ^a^
	100	−31.87 ± 0.97 ^bc^	9.97 ± 0.04	12.33 ± 0.29 ^b^
	1000	−32.27 ± 0.81 ^bc^	9.92 ± 0.03	14.49 ± 0.16 ^c^
	2000	−34.47 ± 0.53 ^d^	9.91 ± 0.05	16.78 ± 0.15 ^d^
Snowtex-ZL (large SiNPs)	10	−31.17 ± 0.29 ^b^	9.92 ± 0.05	110 ± 1.33 ^e^
	100	−33.37 ± 0.88 ^cd^	9.95 ± 0.05	113.57 ± 0.79 ^f^
	1000	−36.67 ± 0.33 ^e^	9.9 ± 0.05	116.03 ± 0.54 ^g^
	2000	−40.4 ± 0.96 ^f^	9.92 ± 0.06	119.87 ± 0.33 ^h^

**Table 2 nanomaterials-13-02110-t002:** Seed germination parameters in tomato seeds treated with suspensions of silica nanoparticles of different sizes and at different concentrations. Different letters indicate significant differences (*p* ≤ 0.05) according to the Duncan test.

Silica Size	Concentration (mg/L)	MGT (Days)	GI	CVG
-	0 (control)	5.24 ± 0.29 ^b^	18.8 ± 1.47 ^b^	19.14 ± 1.01 ^b^
Small SiNPs	10	5.04 ± 0.27 ^ab^	19.8 ± 1.33 ^ab^	19.9 ± 1.07 ^ab^
	100	4.8 ± 0.34 ^a^	21 ± 1.67 ^a^	20.93 ± 1.44 ^ab^
	1000	4.64 ± 0.29 ^a^	21.8 ± 1.47 ^a^	21.64 ± 1.34 ^a^
	2000	5.04 ± 0.08 ^ab^	19.8 ± 0.4 ^ab^	19.85 ± 0.31 ^ab^
Large SiNPs	10	4.92 ± 0.2 ^ab^	20.4 ± 1.02 ^ab^	20.36 ± 0.86 ^ab^
	100	4.68 ± 0.27 ^a^	21.6 ± 1.36 ^a^	21.44 ± 1.23 ^a^
	1000	4.76 ± 0.34 ^a^	21.2 ± 1.72 ^a^	21.12 ± 1.58 ^a^
	2000	4.96 ± 0.29 ^ab^	20.2 ± 1.47 ^ab^	20.24 ± 1.29 ^ab^

**Table 3 nanomaterials-13-02110-t003:** Parameters of seedlings vigor for seedlings germinated and grown in suspensions of articles of different sizes at different concentrations. Different letters indicate significant differences (*p* ≤ 0.05) according to the Duncan test.

Silica Size	Concentration (mg/L)	VI_LENGTH_	VI_FW_	VI_DW_
-	0 (control)	946.4 ± 64.79 ^b^	2.19 ± 0.09 ^c^	0.2 ± 0.01 ^c^
Small SiNPs	10	1082 ± 46.27 ^a^	2.35 ± 0.08 ^abc^	0.21 ± 0.01 ^abc^
	100	1048 ± 68.49 ^ab^	2.32 ± 0.11 ^bc^	0.21 ± 0.01 ^bc^
	1000	1117.2 ± 81.54 ^a^	2.52 ± 0.11 ^a^	0.22 ± 0.01 ^a^
	2000	1061.6 ± 66.49 ^ab^	2.42 ± 0.11 ^ab^	0.22 ± 0.01 ^ab^
Large SiNPs	10	1088.4 ± 101.76 ^a^	2.48 ± 0.18 ^ab^	0.22 ± 0.02 ^ab^
	100	1102 ± 121.36 ^a^	2.52 ± 0.11 ^a^	0.23 ± 0.01 ^a^
	1000	1060.4 ± 83.96 ^ab^	2.46 ± 0.05 ^ab^	0.22 ± 0.01 ^ab^
	2000	1081.2 ± 27.47 ^a^	2.32 ± 0.1 ^bc^	0.21 ± 0.01 ^bc^

## Data Availability

All data generated or analyzed during this work are included in this published article.

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
