# Peer review of "The Role of Silica Nanoparticles in Promoting the Germination of Tomato (Solanum lycopersicum) Seeds"

_nanomaterials, 2023, doi:10.3390/nano13142110_

Round 1
Reviewer 1 Report
The article entitled: “The Role of Silica Nanoparticles in Promoting the Germination of Tomato (Solanum lycopersicum) Seeds” presents a very interesting look at the problem of tomato seeds' growth in the presence of silica nanoparticles. The paper is very interesting, well-written and presents consistent results, and attempts to answer the question of the mechanism behind the observed effect. I think the paper should be published, but after taking into account a few comments and clarifying a few things:
-
Page 2, lines 86-87: On what basis do the authors conclude about the structural, physicochemical properties, size, and stability in water? (This information does not appear until later, which makes it lacking here whether these properties were obtained from the authors' own research or received from the manufacturer).
-
Page 2, lines 86-89: On what basis was it decided to perform tests in an alkaline environment with a pH of 9.5-10.5? Here, there are some inaccuracies because the authors mentioned the stability in water and then present the information about stability in alkaline solution. In my opinion, there are duplicated data and it is worth rewriting this section to make it more consistent.
-
Table 1: The results show that with an increase in the concentration of the small Zeta potential and the size of the nanoparticles increases. Unfortunately, these observations are not addressed in the text. The mechanism/effect behind the observed changes and whether it could have an effect on the estimated seed growth parameters was not pre-discussed. In addition, whether the changes are statistically dependent or not was not discussed.
-
page 5. I don't quite like the approximation of the data as done in line 192 compared to the data in Table 1. The particle size is not 10 nm but around 14 nm for a particular concentration 1000 mg/L and similar for other data. I suggest introducing some abbreviated nomenclature like “small-size nanoparticles or larger nanoparticles” or giving specific values in the text consistent with those as in Table 1.
-
Page 5, lines 186-194: Days with uncertainties are given but not tabulated data for individual MGT, GI, and CVG parameters. It would be worthwhile to add a table with the results.
-
Fig. 2 is unreadable. Is it possible to make larger bars and mark measurement uncertainties?
-
Page 6, line 218: On what basis do the authors spring about "contact area" and "hydrophilicity"? Have tests been done to post-affirm these hypotheses, or have they verified that the nanoparticles are hydrophilic?
-
Fig. 3 Has an SEM-EDS chemical distribution been checked and performed to confirm where the SiO2 nanoparticles are located?
-
Figs. 2 and 4 What do a and b mean?
-
Page 9 line 270, and page 11 lines 359, 370, and 371: italic
The article is interesting but needs some improvements.
Author Response
RESPONSE TO REVIEWER#1
Point 1,2
- Page 2, lines 86-87: On what basis do the authors conclude about the structural, physicochemical properties, size, and stability in water? (This information does not appear until later, which makes it lacking here whether these properties were obtained from the authors’ own research or received from the manufacturer).
- Page 2, lines 86-89: On what basis was it decided to perform tests in an alkaline environment with a pH of 9.5-10.5? Here, there are some inaccuracies because the authors mentioned the stability in water and then present the information about stability in alkaline solution. In my opinion, there are duplicated data and it is worth rewriting this section to make it more consistent.
Response 1, 2:
We apologize for the irregularity of the sentence. The properties of the silica nanoparticles used in the present study were obtained from the manufacturer’s information. The nanoparticles are amorphous, spherical, negatively charged, and have an alkaline pH. We have rewritten the sentences and added related information to provide more information about the colloids.
Lines 87-90: These aqueous nanoparticles are amorphous, spherical [19], negatively charged [20], and stable at an alkaline pH of 9.5–10.5 without added organic surfactants [21]. The nanoparticles were used as received and diluted to the specified concentration for further investigation.
[19] Okubo, T. Determination of the effective charge numbers of colloidal spheres by conductance measurements. J. Colloid Interface Sci. 1988, 125, 380-385.
[20] Matsumoto, T. Influence of ionic strength on rheological properties of concentrated aqueous silica colloids. J. Rheol. 1989, 33, 371-379.
[21] Yano, S.; Maeda, H.; Nakajima, M.; Hagiwara, T.; Sawaguchi, T. Preparation and mechanical properties of bacterial cellulose nanocomposites loaded with silica nanoparticles. Cellulose. 2008, 15, 111-120.
We also measured the characteristics of silica nanoparticles in terms of particle size distribution and zeta potential because these parameters were affected by the different concentrations used. We also provided explanations about this evidence in the Discussion section (Lines 294-308)
The silica nanoparticles we used were stable at an alkaline pH as stated by the manufacturer. Therefore, during our investigations, we always made sure to keep the pH of the nanoparticle suspension within that pH range. This condition would make the colloid become stable.
Point 3:
- Table 1: The results show that with an increase in the concentration of the small Zeta potential and the size of the nanoparticles increases. Unfortunately, these observations are not addressed in the text. The mechanism/effect behind the observed changes and whether it could have an effect on the estimated seed growth parameters was not pre-discussed. In addition, whether the changes are statistically dependent or not was not discussed.
Response 3:
We have added explanations regarding the relationship between differences in concentration and characteristics of silica nanoparticles as follows:
Lines 293-303: Increasing the concentration of SiNPs caused an increase in the value of the zeta potential and particle size distributions measured (Table 1). When nanoparticles were dispersed in a liquid medium, they acquired a surface charge due to the adsorption of ions or functional groups [39]. This surface charge determined the zeta potential. If the surface charge was negative, the zeta potential would be negative, and the nanoparticles would repel each other. The concentration of nanoparticles could affect the zeta potential through factors such as particle-particle interactions or the availability of ions in the suspension [40]. At low concentrations, the electrostatic interactions between nanoparticles might be weaker, leading to less negatively charged (zeta potential) values. Dilution of nanoparticle suspension could potentially result in changes in the particle size distribution [41]. Dilution could disrupt attractive interactions between nanoparticles.
We have also explained how changes in particle size affect seed germination based on the results we obtained from six germination parameters (MGT, GI, CVG, VILENGTH, VIFW, and VIDW).
(Lines 311-322)
The concentration of nanoparticles has been found to have a more pronounced effect on germination compared to the size of nanoparticles as shown by some of the germination parameters in Table 2 and Table 3. For example, a study reported that the particle size of silver nanoparticles had no significant effect on lettuce germination, but the concentration did [42]. Higher nanoparticle concentrations corresponded to an increased total surface area of nanoparticles available for interactions with seeds. This larger surface area provided more opportunities for chemical reactions, adsorption, and physical interactions with seed surfaces, potentially affecting germination-related processes [43]. While nanoparticle size could influence their physicochemical properties and interactions with seeds, the concentration often superseded size effects in terms of their impact on germination. However, it is worth noting that specific responses could still vary depending on the nanoparticle type, seed species, and experimental conditions.
Point 4
- page 5. I don't quite like the approximation of the data as done in line 192 compared to the data in Table 1. The particle size is not 10 nm but around 14 nm for a particular concentration 1000 mg/L and similar for other data. I suggest introducing some abbreviated nomenclature like “small-size nanoparticles or larger nanoparticles” or giving specific values in the text consistent with those as in Table 1.
Response 4:
Thank you for the suggestion. We have changed our terms for SiNPs to “small” (for size of 10 – 17 nm) and “large” (for size of 110 – 120 nm) in the manuscript.
Point 5
- Page 5, lines 186-194: Days with uncertainties are given but not tabulated data for individual MGT, GI, and CVG parameters. It would be worthwhile to add a table with the results.
Response 5:
Thank you for the advice. We have changed the MGT, GI, and CVG data to be tabulated in Table 2 instead of Figure.
Point 6
- Fig. 2 is unreadable. Is it possible to make larger bars and mark measurement uncertainties?
Response 6:
We have converted it to a Table.
Table 2 for MGT, GI, and CVG
Table 3 for VILENGTH, VIFW, and VIDW
Point 7
- Page 6, line 218: On what basis do the authors spring about "contact area" and "hydrophilicity"? Have tests been done to post-affirm these hypotheses, or have they verified that the nanoparticles are hydrophilic?
Response 7:
Silica nanoparticles that we used were hydrophilic. We have added some references.
Lines 220-224:
This finding may also be related to the hydrophilic nature of silica [32,33]. Increasing the contact surface area between SiNPs and seeds could enlarge the hydrophilic space around the seeds and enhance imbibition, thus promoting earlier germination. We believe that this hydrophilic property will affect the interaction of water around the seeds (especially during imbibition) and thereby influence germination.
[32] Kawaguchi, M.; Yamamoto, T.; Kato, T. Rheological studies of hydrophilic and hydrophobic silica suspensions in the presence of adsorbed poly (N-isopropylacrylamide). Langmuir. 1996, 12, 6184-6187.
[33] Shar, J.A.; Obey, T.M.; Cosgrove, T. Adsorption studies of polyethers: part II: adsorption onto hydrophilic surfaces. Colloids Surf. 1999, 150, 15-23.
Point 8
- Fig. 3 Has an SEM-EDS chemical distribution been checked and performed to confirm where the SiO2 nanoparticles are located?
Response 8:
The EDS system which attached in our SEM facilities cannot detect the element of an object smaller than 1 micrometer. Our silica nanoparticles could not be detected via SEM route. Therefore, we used an X-ray fluorescence (XRF) method to quantify Si (of silica nanoparticles) in the seedlings as shown in Figure 3. We also provide explanations on Lines 362-370.
The absence of silica in the seedlings in all treatment groups (Fig. 3) indicated that the silica concentration in the seedlings was probably below the resolution capacity of the XRF spectrometer (< 10 ppm). Silica may only be present on the outside of the seed or in the area around the seed coat (testa). The presence of silica on the outside of the seed coat may have made the conditions around the seeds more hydrophilic and possibly attracted more water to the surface of the seed. This would accelerate the germination process and increase vigour. This is supported by observations from the SEM (Fig. 2) which showed that the surface of the seeds treated with SiNPs became smoother and initiated earlier germination.
Furthermore, we analyzed interactions with Bacillus sp. to better understand the mechanism of nanoparticles during germination. We believe that the presence of nanoparticles will alter the microenvironment (water-interaction) as well as interactions of surrounding microbiome
Point 9
- Figs. 2 and 4 What do a and b mean?
Response 9:
a and b were symbolized to indicate significant differences (p ≤ 0.05) in statistical analysis. We used analysis of variance (ANOVA) to find out if there was a difference in the means. Once we know whether there was a difference, we continued our investigation using post-hoc tests to measure specific differences between the pair means. We used Duncan’s Multiple Range Test (DMRT) at the 0.05 significance level.
We have converted the Figure into a Table.
Table 2 for MGT, GI, and CVG
Table 3 for VILENGTH, VIFW, and VIDW
Point 10
- Page 9 line 270, and page 11 lines 359, 370, and 371: italic
Response 10:
Thank you for the correction. We have changed them.
Reviewer 2 Report
A brief summary
The influence of additions of water-insoluble silica nanoparticles (SiNP) on the germination of tomato seeds in the aquatic environment was studied. Nanoparticles improve seed germination. Nanoparticles with a size of 10 nm stimulate the germination and growth of seeds more effectively than nanoparticles with a size of 110 nm. The combined effect of nanoparticles with Bacillus sp. can have a synergistic positive effect on seed germination as a result of the production of substances that stimulate plant growth. SiNPs nanoparticles are promising as an environmentally friendly additive for improving seed germination in an aquatic environment.
Broad comments
The article is well written, the results and analyzes presented appropriately, the conclusions are substantiated by the results obtained. The article can be published without any edits.
Specific comments
No specific comments - no editing required.
Author Response
RESPONSE TO REVIEWER#2
Point 1
- The article is well written, the results and analyzes presented appropriately, the conclusions are substantiated by the results obtained. The article can be published without any edits.
Response 1:
Thank you for your efforts in reading our manuscript.
Reviewer 3 Report
Comments to the manuscript nanomaterials-2508439 "The Role of Silica Nanoparticles in Promoting the Germination of Tomato (Solanum lycopersicum) Seeds".
Authors propose the report of an experiment of use of nanoparticles of silica to enhance the germination time of tomato seeds. The experiment was correctly designed and the manuscript is well written. The introduction provides a good state of the art and a clear presentation of the research objectives. Materials and methods are correctly presented and results are sound ad clearly showed. The discussion is sufficiently detailed. In my opinion, the manuscript is suitable for publication after minor editing changes.
I suggest the authors to add to the discussion an evaluation of the quantitatively limited consistency of the results in terms of time of seed germination, and to propose future research in order to improve the treatment performance.
The English language is sufficently correct and minor editing changes are required.
Author Response
RESPONSE TO REVIEWER#3
Point 1
- I suggest the authors to add to the discussion an evaluation of the quantitatively limited consistency of the results in terms of time of seed germination, and to propose future research in order to improve the treatment performance.
Response 1:
Thank you for your suggestions. We have added some discussion related to the limited consistency of quantitatively results and how to manage them on Lines 401-410.
The consistency of the results of seed research was influenced by several factors such as genetic variability and environmental conditions. Therefore, standardized protocols, maintaining consistent environmental conditions, and carefully selecting and handling seeds are essential. Good statistical analysis is also necessary to draw appropriate conclusions for seed and other biological research.
Lines 401-410:
The observed germination times for a particular species of seed exhibited variability within a certain range. Even within a single species, there could be genetic variability among seeds. Different individuals within the same species might have slight genetic variations, leading to variations in germination characteristics, including the time it took for seeds to germinate [60]. This genetic diversity could contribute to the inconsistent germination times observed. To address this issue, we can take measures to minimize variability by using standardized protocols, maintaining consistent environmental conditions, and carefully selecting and handling seeds [61]. Conducting multiple replicates and statistical analyses could help identify trends and patterns within the variability and provide a clearer understanding of the average germination time for a particular species [62]
We also proposed some perspectives on future research on nanoparticles and seed germination as listed in Lines 416–430.
Future research on seed germination using nanoparticles can focus on various aspects to further our understanding of their effects. Investigating the dose-response relationships of nanoparticles on seed germination can provide insights into their toxicity thresholds or optimal concentrations. This study area can help establish guidelines for safe nanoparticle application in agriculture and highlight potential risks associated with high concentrations. Understanding the underlying mechanisms through which nanoparticles influence seed germination is crucial. Future research can focus on elucidating the cellular and molecular processes affected by nanoparticles, such as hormone regulation, gene expression, oxidative stress, and nutrient uptake. Advanced techniques like transcriptomics, proteomics, and metabolomics can provide valuable data in this regard. Nanoparticles can interact with the microbial communities associated with seeds, influencing seed germination outcomes. Future research can explore the effects of nanoparticles on the composition and activity of seed-associated microbiota and how these interactions impact germination. This area of study can shed light on the potential indirect effects of nanoparticles on seed germination through their influence on the microbiome.
Round 2
Reviewer 1 Report
Thank you for the responses.